# Antiviral Activity of Chitosan Nanoparticles and Chitosan Silver Nanocomposites against Alfalfa Mosaic Virus

**DOI:** 10.3390/polym15132961

**Published:** 2023-07-06

**Authors:** Sherif Mohamed El-Ganainy, Ahmed M. Soliman, Ahmed Mahmoud Ismail, Muhammad Naeem Sattar, Khaled Yehia Farroh, Radwa M. Shafie

**Affiliations:** 1Department of Arid Land Agriculture, College of Agriculture and Food Sciences, King Faisal University, P.O. Box 420, Al-Ahsa 31982, Saudi Arabia; 2Pests and Plant Diseases Unit, College of Agriculture and Food Sciences, King Faisal University, P.O. Box 420, Al-Ahsa 31982, Saudi Arabia; 3Vegetable Diseases Research Department, Plant Pathology Research Institute, Agricultural Research Center (ARC), Giza 12619, Egypt; 4Virus and Phytoplasma Research Department, Plant Pathology Research Institute, Agricultural Research Center (ARC), Giza 12619, Egypt; ahmedsoliman@arc.sci.eg (A.M.S.);; 5Central Laboratories, P.O. Box 420, Al-Ahsa 31982, Saudi Arabia; 6Nanotechnology and Advanced Materials Central Lab., Regional Center for Food and Feed, Agricultural Research Center (ARC), Giza 12619, Egypt

**Keywords:** *Capsicum annum*, RT-PCR AMV, chitosan, direct ELISA

## Abstract

Plant viruses are a global concern for sustainable crop production. Among the currently available antiviral approaches, nanotechnology has been overwhelmingly playing an effective role in circumventing plant viruses. Alfalfa mosaic virus (AMV) was isolated and identified from symptomatic pepper plants in Egypt using symptomatology, serological tests using the direct ELISA technique, differential hosts and electron microscopy. The virus was biologically purified from a single local lesion that developed on *Chenopodium amaranticolor.* The AMV infection was further confirmed using an AMV coat protein-specific primer RT-PCR. We further evaluated the antiviral potential of chitosan nanoparticles (CS-NPs) and chitosan silver nanocomposites (CS-Ag NC) in different concentrations against AMV infections in pepper plants. All tested concentrations of CS-NPs and CS-Ag NC induced the inhibition of AMV systemically infected pepper plants when applied 24 h after virus inoculation. The foliar application of 400 ppm CS-NPs or 200 ppm CS-Ag NC produced the highest AMV inhibitory effect (90 and 91%) when applied 24 h after virus inoculation. Treatment with CS-NPs and CS-Ag NC considerably increased the phenol, proline and capsaicin contents compared to the infected plants. Moreover, the agronomic metrics (plant height, fresh and dry pod weights and number of pods per plant) were also significantly improved. According to our results, the potential applications of CS-NPs and CS-Ag NC may provide an effective therapeutic measure for better AMV and other related plant virus management.

## 1. Introduction

Hot peppers (*Capsicum annum*), a member of the *Solanaceae* family, are considered an essential and seasoning vegetable due to their distinctive taste, color, flavor, spice, aroma and medicinal value. Furthermore, they are a good source of antioxidants, including phenols, flavonoids, anthocyanin, vitamins, B-carotene and capsaicin, which are involved in defense mechanisms and are important for human health [1]. Viral diseases are one of the main threats to pepper production, causing significant economic losses by reducing the quality and quantity of the marketable yield [2]. Peppers have been known to host diverse plant viruses of the *Bromoviridae*, *Bunyaviridae*, *Geminiviridae*, *Luteoviridae*, *Potyviridae* and *Virgaviridae* families. Among these viruses, alfalfa mosaic virus (AMV) (family *Bromoviridae*, genus *Alfamovirus*) is an economically significant and widespread virus in Egypt [3]. It has been reported in Australia, France, Greece, Iran, Italy, New Zealand, Saudi Arabia and North America [4]. Besides the main crop hosts of the Fabaceae and Solanaceae families, AMV has a wide host range, infecting ~430 herbaceous and woody plant species in 51 families with a wide range of symptoms [5]. Aphids play a vital role in the biological transmission of AMV non-persistently [6]. In addition, AMV can also be transmitted mechanically or through seeds [7]. Moreover, uncultivated plants and weeds are also crucial for the prevalence and persistence of AMV as secondary plant hosts [8].

The viral architecture of AMV is built on icosahedral particles 30–57 nm in length and a diameter of 18 nm. The RNA-based AMV genome is divided into three single-stranded positive-sense RNA components (RNA1, RNA2 and RNA3). The replicase subunits P1 and P2 are encoded on RNA1 and RNA2 molecules, respectively. The movement protein (MP) and coat protein (CP) are encoded on the RNA3 molecule. The CP is translated via the additional sub-genomic component RNA4 and is critical in the translation and binding of RNAs, viral shuttling in the nucleus and cytoplasm, virion assembly and systemic viral movements [9].

Plant viruses are among the most critical pathogens jeopardizing global agriculture and food security. Using ecofriendly antiviral agents can be an alternate approach to elicit a plant innate resistance development against viruses. Chitosan is a natural biopolymer with enormous biological benefits due to its unique characteristics, positive charge and the presence of reactive hydroxyl and amino groups. Due to its bioadhesive, biodegradable and biocompatible properties, chitosan is a safe and cost-effective biocontrol agent [10]. Although the antiviral activity of chitosan nanoparticles (or their derivatives) has been shown in some studies, its full potential has yet to be explored. For example, silver nanoparticles (AgNPs) used in controlling viral infections are still limited. The antiviral activity of AgNPs/chitosan composites has been tested against the influenza A virus. It was shown that the inhibitory effect against H1N1 improved as the concentration of AgNPs increased [11]. Chitosan/dextran nanoparticles (CDNPs) at a conc. of 100 mg L^−1^ reduced the disease severity of AMV in treated *Nicotiana glutinosa* plants [12]. Similarly, chitosan nanoparticles at a conc. of 300 and 400 mg/L inhibited bean yellow mosaic virus (BYMV) infection and substantially reduced viral accumulation [13].

Pepper (*Capsicum annum* L.) plants treated with chitosan silver nanoparticles (CS-Ag NPs) and chitosan nanoparticles (CS-NPs) significantly increased the vegetative plant growth and the total phenol, proline and capsaicin accumulation [14,15]. The use of chitosan nanoparticles as a biostimulant has been explored in many crop plants [16,17]. Nevertheless, studies on the antiviral activities of chitosan (particularly CS-AgNPs) are still limited. Therefore, this study was designed to synthesize and characterize CS-AgNPs and CS-Ag NC to explore their role in inducing plant defenses in pepper plants against AMV infection.

## 2. Materials and Methods

### 2.1. Virus Isolation and Identification

Ten pepper (*Capsicum annum* L.) plant samples with suspected AMV-like symptoms were collected from a pepper field in the Ismailia Governorate, Egypt. The observed symptoms included yellow mosaic on the leaves, mottling, curling, chlorosis, yellow blotching and chlorotic sectors. The isolated virus was maintained on ten pepper plants for the propagation of the virus and used as a source for subsequent studies. The leaf samples of healthy and symptomatic pepper plants were tested for the presence of the known viruses from peppers, viz., the potato virus Y (PVY), cucumber mosaic virus (CMV), AMV and tomato spotted wilt virus (TSWV), using the Double Antibody Sandwich-Enzyme-Linked Immunosorbent Assay (DAS-ELISA), as described earlier [18]. PVY, CMV and AMV were tested using the LOEWE Biochemica GmbH Kit (Biochemica, Mühlweg, Germany), while TSWV was tested using the Bioreba kit (Bioreba, Reinach, Switzerland). The positive samples for AMV were used as a source of virus inoculum, and the diagnostic host plants were mechanically inoculated: *Phaseolus vulgaris*, *Vigna unguculata*, *Vicia faba*, *Datura stramonium*, *Nicotiana glutinosa*, *Chenopodium amaranticolor*, *C. quinoa*, *C*. *murlae*, *Catharanthus roseus* and *Ocimum basilicum*. For biological purification, *C. amaranticolor* plants grown under natural lighting with day/night temperatures of approx. (23 ± 2 °C) were inoculated against the virus as a local lesion host for three consecutive passages [19], then transmitted mechanically to pepper plants.

### 2.2. RNA Isolation and Reverse Transcription Polymerase Chain Reaction RT-PCR Amplification of AMV Coat Protein Gene

According to the manufacturer’s manual, the total RNAs were extracted from newly emerged leaves of naturally infected, mechanically inoculated and healthy pepper plants using a Gene jet^TM^ plant RNA purification mini kit (Thermo Fisher Scientific, Carlsbad, CA, USA). RT-PCR was conducted using the verso^TM^ one-step RT-PCR kit (Thermo Fisher Scientific Co., Waltham, MA, USA), according to the manufacturer’s instructions. The oligonucleotide primers AMVcoat-F/AMVcoat-R [20] were used to amplify the conserved region in the CP gene of AMV by one-step RT-PCR. The RT-PCR reaction was optimized to be performed in a final volume of 25 µL containing 3 µL RNA (4 ng/µL), 12.5 µL of one-step PCR master mix (2×), 3 µL of 10 µM of each primer, 0.5 µL of Verso enzyme mix, 1.25 µL of RTEnhancer and 4.75 µL of nuclease-free water. Amplification was performed in an automated T Gradient Biometra (Biometra, Jena, Germany) thermal cycler. Samples were amplified using the following cycling parameters: hold at 50 °C for 30 min (RT step) and hold at 95 °C for 15 min (hot start to PCR); the tubes were heated at 94 °C for 2 min and then subjected to 35 cycles of amplification: 30 s at 94 °C for denaturation, 30 s at 58 °C for annealing and 30 s at 74 °C for extension, followed by a final hold at 72 °C for 10 min. RT-PCR products were analyzed by 1% agarose gel electrophoresis, stained with gel star (Lonza, Carlsbad, CA, USA) and visualized by UV illumination (Gel Doc 2000, Bio-Rad, Carlsbad, CA, USA). The DNA ladder (1 kb) was used for comparison.

### 2.3. Preparation of Chitosan Nanoparticles (CS-NPs)

Chitosan nanoparticles (CS-NPs) were prepared by the ionic gelation method according to Calvo et al. [21]. The technique utilized the electrostatic interaction between the amine group of chitosan and a negatively charged group of polyanion, such as trisodium polyphosphate (TPP) and CS aqueous solution (0.2% *w*/*v*) (molecular weight 50,000–190,000 Da, degree of deacetylation 75–85% and viscosity: 20–300 cP, Sigma-Aldrich, Missouri, USA), prepared by dissolving CS in acetic acid solution (1% *v*/*v*) (99–100%, Riedel-de Haën) at room temperature. Subsequently, the TPP solution (0.06% *w*/*v*) (Sigma-Aldrich, St. Louis, MO, USA) was added dropwise to the CS solution under vigorous stirring for 30 min. The CS-NPs suspension was freeze-dried before further use or analysis.

### 2.4. Preparation of Chitosan Silver Nanocomposites (CS-Ag NC)

Chitosan silver nanocomposites (CS-Ag NC) were prepared by the chitosan reduction of silver nitrate, according to Babu et al. [22]. The polymeric chain’s amino groups coordinated the silver ions in an acidic chitosan solution. Ion reduction to metallic silver nanoparticles was coupled with the chitosan hydroxyl group’s oxidation. Briefly, chitosan aqueous solution (1% *w*/*v*) (molecular weight 50,000–190,000 Da, degree of deacetylation 75–85% and viscosity: 20–300 cP, Sigma-Aldrich, St. Louis, MO, USA) was prepared by dissolving chitosan in an acetic acid solution (1% *v*/*v*) (99–100%, Riedel-de Haën, Chapel Hill, NC, USA) at room temperature. Subsequently, the silver nitrate solution (0.01 M) (Sigma-Aldrich, St. Louis, MO, USA) was added immediately into the suspension under continuous stirring for two hours. Sodium borohydride (20 mL, 0.04 M) (Sigma-Aldrich, St. Louis, MO, USA) was added to the previous suspension, and an immediate color change from pale yellow to brown was observed. The resulting CS-Ag NC suspension was centrifuged at 20,000× *g* for 30 min. The pellet was resuspended in deionized water. The CS-Ag NC suspension was freeze-dried before further use or analysis.

### 2.5. Characterization of Chitosan Nanoparticles and Chitosan Silver Nanocomposites

The exact morphology of the prepared CS-NPs and CS-Ag NC was examined using a high-resolution transmission electron microscope (HR-TEM) JEOL (JEM-1400 TEM, Tokyo, Japan) operating at an accelerating voltage of 200 kV (Tecnai G2, FEI, Amsterdam, The Netherlands). The diluted CS-Ag NC solution was ultra-sonicated for 5 min to reduce particle aggregation. Three droplets of the sonicated solution were applied with a micropipette on a copper grid coated with carbon, and they were then allowed to dry at room temperature. The HR-TEM images of the CS-NPs and CS-Ag NC deposited on the grid were captured for morphological evaluation. Using photon correlation, the nanoparticles’ size (Z-average mean) and zeta potential were examined using zetasizer 3000 HS to perform the spectroscopy and a laser Doppler anemometer in triplicate, respectively (Malvern Instruments, Zs Nano, Almelo, The Netherlands). The chemical structure of the prepared CS-NPs and CS-Ag NC was assessed using the X-ray diffraction (XRD) technique. The corresponding XRD pattern was recorded in the scanning mode (X ‘pert PRO, PAN analytical, Almelo, The Netherlands) operated by a Cu K radiation tube (=1.54 A˚) at 40 kV and 30 mA. The default ICCD library in PDF4 was used to analyze the resulting diffraction pattern. Qualitative and quantitative measurements of the applied silver concentrations in CS-Ag NC and a sample of pepper fruits after drying were determined by the inductivity coupled plasma (ICP) technique (PerkinElmer ICP-OES: Optima 2000, Rodgau, Germany). The synthesis and characterization of the CS-NPs and CS-Ag NC were performed in the Nanotechnology & Advanced Materials Central Laboratory (NAMCL), Agricultural Research Center, Egypt.

### 2.6. Effect of Foliar Application of Chitosan Nanoparticles and Chitosan Silver Nanocomposites on Virus Infectivity and Plant Vegetative Growth

The effect of CS-NPs at conc. of 400, 200, 150 and 100 ppm and CS-Ag NC at conc. of 200, 150, 100 and 50 ppm in controlling AMV was evaluated on pepper plant seedlings to prevent AMV infection under greenhouse conditions. Pepper seeds were grown in plastic pots packed with sand and clay (1:2 *v*/*v*) and under natural lighting and in day/night temperatures of approx. (23 ± 2 °C). After three weeks of growth, the seedlings were transferred to 40 × 40 cm pots (4 seedlings per pot). The seedlings were divided into four groups. In group I, the seedlings were first treated with CS-NP or CS-Ag NC and then mechanically inoculated with AMV inoculum (1 mL/plant) after 24 h. In group II, the seedlings were first inoculated with AMV inoculum and then treated with CS-NPs or CS-Ag NC after 24 h. In group III, the seedlings were treated either with CS-NPs or CS-Ag NC immediately after virus inoculation. In group IV (the control group), the seedlings were sub-grouped as C1—seedlings inoculated only with the AMV inoculum (positive control), C2—untreated seedlings (healthy control), C3—seedlings treated with CS-AgNC and C4—seedlings treated with CS-NPs. Four plant seedlings were used for each treatment in each group and sub-group. All the plants received the recommended agronomic practices and were observed routinely for symptom development. The virus inhibition percentage (%) of the tested plants was determined through DAS-ELISA three weeks after inoculation, as described by Devi et al. [23] using the following equation: Inhibition % = (A − B/A) × 100, where A is the number of plants in the positive control experiment (untreated), and B is the number of treated inoculated plants. The newly emerged plant leaves were used for RNA extraction and RT-PCR analysis, as described above.

Plant height (cm), fresh and dry weights of pods (g) and number of pods per plant were recorded 7 months after planting.

### 2.7. Effect of Various Concentrations of Chitosan Nanoparticles and Chitosan Silver Nanocomposites on Active Ingredients in Pepper Pods

The determination of capsaicin was achieved according to Peng and Wang [24] as mg/kg DW. The determination of the proline content was achieved according to Bates et al. (1973) as mg gFW^−1^.

The determination of the total phenols in the pods was conducted according to the method described by Diaz and Martin [25] as (mg/100 g of fresh matter).

### 2.8. Data Analysis

The mean value was calculated by performing 2-way or 1-way analyses of variance (ANOVA) using Statgraphics Centurion XVI (Statpoint Technologies, Inc., Warrenton, VA, USA). When appropriate, the means were separated by Fisher’s Protected LSD test (*p* < 0.05) [26].

## 3. Results

### 3.1. Collection of Field-Infected Pepper Plants and Preparation of Virus Inoculum

The symptomatic pepper plants showed yellow mosaic on leaves, mottling, curling, chlorosis, yellow blotching and chlorotic sectors symptoms compared to the non-symptomatic plants (Figure 1A,B). The young leaf samples were detached from each plant and tested for the presence of common pepper-infecting plant viruses using DAS-ELISA and RT-PCR (Appendix A). The results showed that all ten plants tested negative for the presence of PVY, CMV and TSWV, while these were positive for only AMV infection.

For further propagation, the sap from each pepper plant was inoculated onto ten pepper plants under greenhouse conditions (Figure 1C–E). The plants started showing typical AMV symptoms 21 days post-inoculation (DPI). All the plants were tested for the presence/absence of AMV using DAS-ELISA and RT-PCR. It was found that all the plants tested positive for the presence of AMV.

The AMV inoculum from pepper plants was used to inoculate ten differential host plant species belonging to six families (Appendix A). The symptoms induced in the tested hosts ranged between mosaic, chlorotic and necrotic local lesions; necrotic spots and yellowing (Appendix A and Table 1). The results grouped all differential host plants into two groups. The plants *C*. *amaranticolor*, *C*. *quinoa*, *C*. *mural*, *Phaseolus vulgaris* and *Vigna unguiculata* were grouped together to show only the local lesions that developed 5–7 DPI, whereas *Catharanthus roseus*, *Vicia faba*, *Ocimum basilicum*, *Datura stramonium* and *Nicotiana tabacum* were grouped separately for showing systemic infections 21 DPI. All plants were tested with DAS-ELISA and RT-PCR for AMV infection.

### 3.2. Characterization of Chitosan Nanoparticles and Chitosan Silver Nanocomposites

The synthesized CS-NPs and CS-Ag NC were characterized for physiochemical properties using HR-TEM (Figure 2A,B). The CS-NPs had a nearly spherical shape, smooth surface and size range of about 30 nm. The hydrodynamic diameter of the CS-NPs was 37.8 nm, with a zeta potential +48.4 mV (Figure 2A). The crosslinking between chitosan and silver in CS-Ag NCs was spherical in shape, with a smooth surface and size range of about 11.33 nm. The hydrodynamic diameter of CS-Ag NC was 12.55 nm, with a zeta potential +65.1 mV (Figure 2B). The X-Ray diffraction patterns of CS-NPs and CS-Ag NC were also determined (Figure 2A,B). The XRD pattern of the CS-NPs showed a broad typical hump peak start from 2θ = 10° to 2θ = 30°. The main peak of the CS-NPs pattern was observed at 2θ = 20°. The peaks at 2θ = 38.13°, 64.46° and 77.42° were assigned to (111), (220) and (311) of CS-Ag NC.

### 3.3. Effect of Chitosan Nanoparticles and Chitosan Silver Nanocomposites on Virus Infectivity

Based on the DAS-ELISA analysis, the treated pepper plant seedlings showed significant variations in inhibiting viral proliferation compared to the control treatments (Table 2). However, spraying 400 ppm CS-NPs and 200 ppm CS-Ag NC on all groups showed the most significant results. The inhibitory effect of 200 ppm CS-Ag NC and 400 ppm CS-NPs in group II showed the highest inhibitory effect against AMV infection in pepper plants (91% and 90%). Similarly, in group III, 200 ppm CS-Ag NC and 400 ppm CS-NPs produced a higher inhibitory effect (78% and 76%, respectively) than all the other treatments. In addition, the seedlings treated with 200 ppm CS-Ag NC and 400 ppm CS-NPs in group I also showed significantly high inhibitory effects (67% and 60%, respectively) compared to the control treatments. However, amongst all the treatments, the 100 ppm CS-NPs pre-inoculation treatment showed the lowest inhibitory effect (47%; Table 2).

### 3.4. Effect of Various Concentrations of Chitosan Silver Nanocomposites and Chitosan Nanoparticles on Active Ingredients in Pepper Pods

The results (Table 3) showed an increase in the total phenols in all treatments compared to the healthy control. The highest significant phenol contents were produced in group II pepper plant pods treated with 200 ppm CS-Ag NC (1.83) and 400 ppm CS-NPs (1.80), respectively, whereas the pepper plants in group I treated with 200 ppm CS-Ag NC and 400 ppm CS-NPs produced low phenol contents (1.62 and 1.58), respectively. The plants infected with AMV gave a higher value of the phenol content (1.22) than the healthy control (1.12).

The capsaicin content was recorded as the highest in the healthy control treatments (622.17) compared to the AMV-positive control (369.45), whereas the capsaicin content was improved in the plants treated with 400 ppm CS-NPs and 200 ppm CS-Ag NC in all the treatment groups. It was found that, in group II, the treatment of 200 ppm CS-Ag NC and 400 ppm CS-NPs increased the capsaicin content to 481.79 and 478.83, respectively (Figure 3). Similarly, the treatment of plants with 400 ppm CS-NPs and 200 ppm CS-Ag NC in group III also improved the capsaicin content to 200 ppm CS-Ag NC and 472.33 and 470.47, respectively. Similarly, in group I, the capsaicin content was also enhanced in plants treated with 200 ppm CS-Ag NC (401.20) and 400 ppm CS-NPs (390.73), respectively.

The data also showed that AMV infection increased the proline content (0.95) in pods compared to the healthy control (0.75). The proline content was significantly highest in plants treated with 200 ppm CS-Ag NC (1.23) and 400 ppm CS-NPs (1.20) in group II compared to the other concentrations. The proline contents were recorded as 1.17 and 1.16 in plants treated with 200 ppm CS-Ag NC and 400 ppm CS-NPs in group III, while, in group I, the treatments with 200 ppm CS-Ag NC and 400 ppm CS-NPs produced the lowest values for the proline content (1.14 and 1.13), respectively.

### 3.5. Effect of Chitosan Silver Nanocomposites and Chitosan Nanoparticles on Growth and Yield of Pepper Plants Inoculated with AMV

The results showed that AMV infection significantly affected the vegetative metrics (Table 4). A significant reduction was observed in plant height (78.1%), fresh (54.0%) and dry weights of pods (35.4%) and number of pods per plant (57.28%) for AMV-infected plants compared to the untreated healthy controls. Amongst all the treatments, the treatments with 400 ppm CS-NPs and 200 ppm CS-Ag NC in group II significantly improved all the vegetative parameters compared to the AMV-infected plants. It was found that 200 ppm CS-Ag NC and 400 ppm CS-NPs showed significantly increased plant heights (60.6 cm and 58.9 cm), fresh weights (21.2 g and 20.6 g) and dry weights of pods (10.3 g and 10.1 g ) and number of pods per plant (24.7 and 23.5), respectively.

## 4. Discussion

Plant viral infections have a significant negative economic impact on sustainable agriculture. Among various cutting-edge approaches, using nanoparticles has been proven as a novel approach to withstand viral infections in different crop plants [27,28,29]. AMV is an emerging threat to crop production in Egypt [30,31]. During a survey in the Ismailia Governorate of Egypt, symptomatic pepper plants showed typical AMV symptoms, as previously described [31]. RT-PCR produced an amplicon size of ~700 bp, equivalent to the AMV CP gene. AMV CP is the most critical region in the viral genome and is an essential criterion for describing the identification and taxonomy of the virus [32]. After confirmation, the field-collected plants were maintained in a greenhouse and were used as a source for AMV inoculum to inoculate ten differential plant hosts representing six different plant families. Based on the type of symptoms and infection, these plants were grouped into two major categories, i.e., those developing only local lesions and others showing systemic infections as well. The results showed that AMV can potentially threaten *C. roseus*, *V. faba*, *O. basilicum*, *D. stramonium* and *N. tabacum* hosts, because these plants showed a systemic development of the symptoms. The virus was easily transmitted mechanically to other plant hosts, as has been previously reported [20,31,33]. AMV is known to cause infections in broad host plants of *Solanaceae* and *Leguminosae* families in Egypt, India, Turkey and Saudi Arabia [34,35]. As previously described [12], *C. amaranticolor* leaves showed single local lesion development in 5–7 DPI and were further used for biological purification through three consecutive passages [19], then transmitted mechanically to pepper plants for further antiviral studies.

Chitosan is a natural polymer used to synthesize nanoparticles and is generally considered safe by the US Food and Drug Administration (U.S. FDA). It has been a new therapeutic method in controlling human [36] and plant infection viruses [12,37,38]. Our study investigated the antiviral ability of CS-NPs and their silver nanocomposites (CS-Ag NCs) against AMV infection in peppers. The TEM analysis revealed that the synthesized CS-NPs were spherical, with a smooth surface and ~30 nm in size, while the hydrodynamic diameter of the CS-NPs was 37.8 nm. In the CS-Ag NCs, the crosslinking between chitosan and silver was spherical in shape, with a smooth surface and size range of 11.3 nm and a hydrodynamic diameter of 12.55 nm. The X-ray diffraction of the CS-Ag NCs indicated that the crystalline structure of synthesized CS-Ag NC presented a cubic-phase structure of silver (JCPDS 04-004-8730). No silver residues were found in the pepper fruit samples treated with CS-NPs and CS-Ag NCs.

The foliar application of CS-NPs and CS-Ag NCs significantly improved plant immunity to inhibit viral proliferation and disease severity. Among all the treatments, 400 ppm CS-NPs and 200 ppm CS-Ag NC applied after 24 h of AMV inoculation showed the most significant results for virus inhibition. Chitosan nanoparticles are known to enhance plant growth by stimulating nutrient uptake, photosynthesis, cell division and the production of plant hormones [39]. These physiological processes are involved in plants’ defense against plant pathogens [37]. Therefore, it is likely that CS-NPs and CS-Ag NCs helped pepper plants to withstand AMV infection in this study. According to the previous studies, chitosan nanoparticles have the potential to attach to virus particles, inhibit viral replication inside infected cells and boost plant immunity and antioxidant defense systems [40,41]. Chitosan increases plant resistance by increasing the activity of ribonucleases and proteases [40]. In turn, it makes it difficult for viruses to efficiently invade plant cells. These findings suggest that chitosan or chitosan-based nanoparticles could be a promising tool to control plant viruses. Nevertheless, the antiviral activity of CS-Ag NCs has been confirmed against human viruses, such as the H1N1 influenza A virus [11]; however, based on our knowledge, this is the first comprehensive study on the antiviral potential of CS-Ag NCs against plant viruses. Our results are per El Gamal et al. [13], who showed that using 300 and 400 mg/L of chitosan nanoparticles after 48 h of bean yellow mosaic virus infection entirely inhibited the viral infection in beans. Furthermore, Abdelkhalek et al. [12] also found that using chitosan/dextran nanoparticles significantly inactivated and suppressed the AMV accumulation in *N. glutinosa* plants by promoting plant growth. Contrary to our results, Abdelkhalek et al. [12] found that the application of nanoparticles 24 h after AMV inoculation (curative treatment) produced the least significant AMV inhibition compared to the protective treatment (24 h pre-inoculation of AMV). Such differences can be related to the formulation of nanoparticles and/or their derivatives.

Hot peppers^’^ active component, capsaicin, is responsible for their hotness and spiciness. The AMV infection significantly reduced the capsaicin content compared to healthy pepper plants. However, using 400 ppm CS-NPs and 200 ppm CS-Ag NC resulted in a significant increase in the capsaicin content when applied after 24 h of AMV inoculation. These results were per those obtained by El-Shazly et al. [42]. They found that tomato plants infected with (TSWV) had reduced levels of lycopene compared to healthy plants. This may be due to the ability of the virus to affect fruit metabolism [43]. The improved capsaicin content was the effect of CS-NPs and CS-Ag NC to circumvent the deleterious effects of AMV and increase the secondary metabolites in the infected pepper plants. Proline has been shown to play a critical protective role in plant cells under abiotic or biotic stress. The proline metabolic pathway was suggested to play a regulatory role in oxidation–reduction balance and cell survival, which may help to explain these effects [44]. The production of phenols was reported to be enhanced in both virus-infected plants and plants treated with nanoparticles, which may function as an antioxidant to scavenge ROS (reactive oxygen species) and increase the quantities of the antioxidants in plant tissues [45,46]. Biotic and abiotic stresses may cause an increase in phenolic metabolism and antioxidant capacity, which may cause an increase in phenolic compounds. Treatments with CS-AgNC and CS-NPs still produce higher phenols and proline contents, because these nanoparticles improve plant resistance to abiotic stress through mechanisms such as (1) triggering plant cell signals due to the increased production of ROS and/or reactive nitrogen species (RNS) and (2) stimulating the plant protection mechanism, which includes enzymatic and non-enzymatic antioxidants [47]. Phenolic compounds can scavenge ROS and prevent cellular oxidation. The higher proline amount might be attributed to a decrease in proline oxidation to glutamate, an increase in protein turnover or a reduction in protein usage. Furthermore, it may activate the proline biosynthesis genes that produce proline from glutamate [48].

Applying chitosan-based nanoparticles has been shown to affect vegetative growth and yield positively and improve plant mineral contents [49,50,51]. Similarly, the foliar application of silver nanoparticles improved the vegetative growth in plants due to increased photosynthetic pigments, indole acetic acid and stimulated protein and carbohydrate biosynthesis pathways [52,53]. Per the previous studies, it was found that applying 400 ppm CS-NPs and 200 ppm CS-Ag NC 24 h post-inoculation improved the plant height, fresh and dry weights of pods and number of pods per plant significantly. The vegetative metrics of the inoculated pepper plants were negatively affected by AMV infection, as shown in various studies [31,54]. However, applying 400 ppm CS-NPs and 200 ppm CS-Ag NC significantly improved the plant height, fresh and dry weights of pods and number of pods per plant compared to all other treatments during this study.

A probable route of action of silver nanoparticles is the cell periphery, where they commence their antiviral activity after their first contact with the glycoprotein of the virus surface [55]. Due to their smaller size, they enter the host plant cells, disrupting the cellular factors and/or viral proteins that aid in viral replication. They disrupt the viral polymerase activity (in RNA viruses) and prevent the development of new virions [56,57].

It is hypothesized that chitosan nanoparticles may have a robust bioreactivity to bind viral RNA, which has negatively charged phosphate groups in its main chain [58,59]. Furthermore, the positively charged nanoparticles could also target the virus coat protein, because all viral proteins contain a negatively charged cluster of glycol proteins. It further supports our speculation that the role of CS-NPs in controlling plant viruses may be substantially influenced by their nanophysiochemical characteristics, particle chemical nature and bioreactivity, where nano-sized chitosan is essential for their antiviral capabilities.

## 5. Conclusions

The current study empirically evaluated the antiviral potential of synthesized CS-NPs and CS-Ag NC in AMV-infected pepper plants. The electron microscopy analysis revealed that the CS-NPs and CS-Ag NC particles were uniform in size and spherical in shape. It was found that, among all the treatments, the foliar application of 400 ppm CS-NPs and 200 ppm CS-Ag NC was significantly effective in controlling AMV infection. Thus, the application of 400 ppm CS-NPs and 200 ppm CS-Ag NC can provide a long-term and viable control not only for AMV but other similar viral infections in crop plants. However, other parameters associated with these nanoparticles and/or their derivatives, viz., size, effective concentration, biocompatibility and environmental safety (if any), must be investigated explicitly before considering their many applications in crop plants.

## Figures and Tables

**Figure 1 polymers-15-02961-f001:**
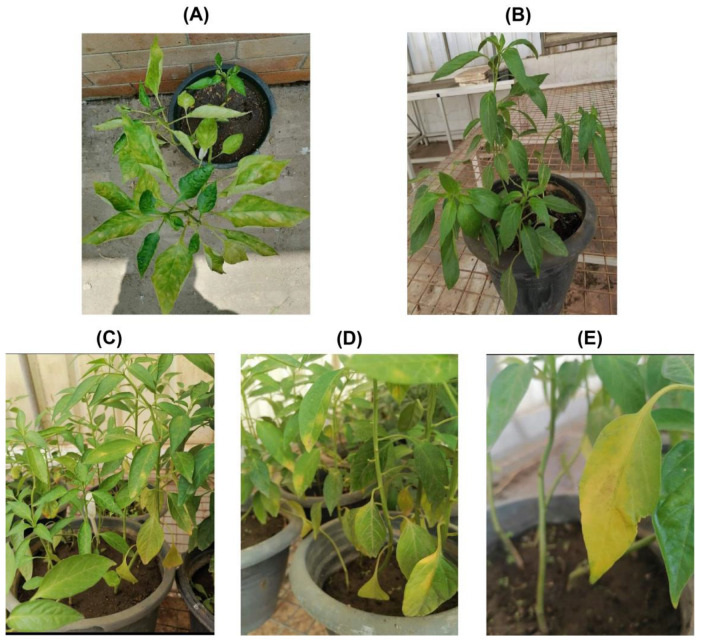
Field-infected symptomatic (**A**) and symptomless or healthy pepper plants (**B**). The sap from symptomatic AMV-infected pepper plants was used to mechanically inoculate greenhouse-grown pepper plants for further confirmation. The mechanically inoculated pepper plants were showing typical AMV symptoms of mosaic (**C**) and leaf curling, yellow blotching (**D**) and chlorosis symptoms (**E**).

**Figure 2 polymers-15-02961-f002:**
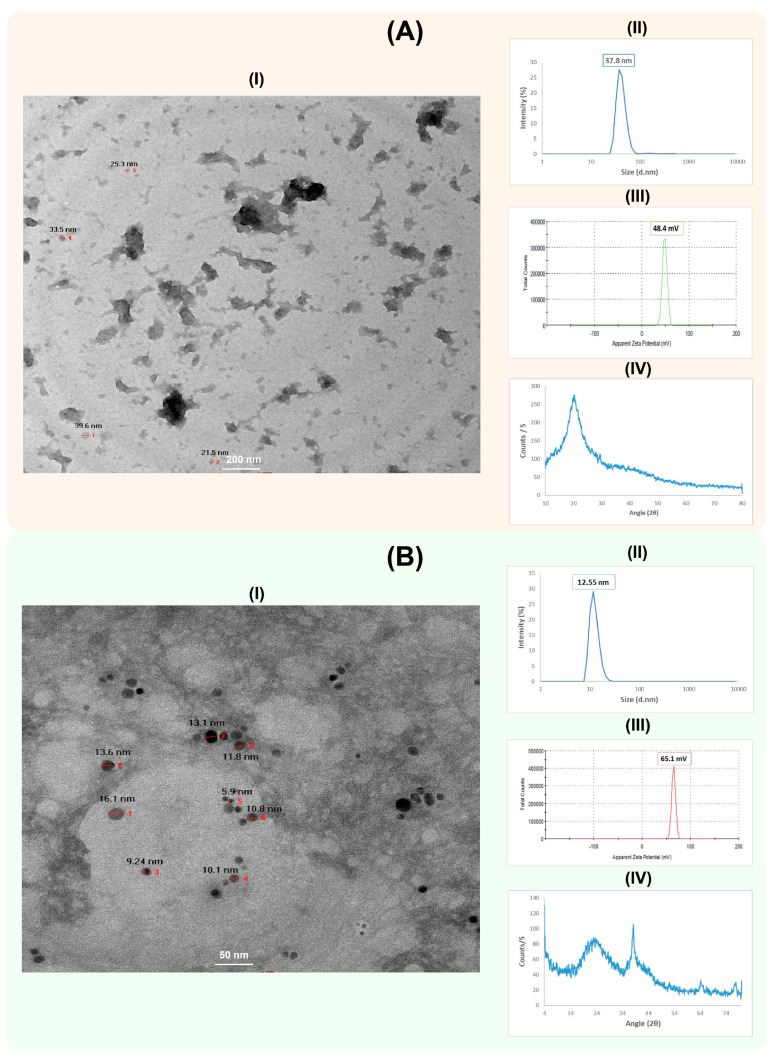
Characterization of chitosan nanoparticles (CS-NPs) (**A**) and chitosan silver nanocomposites (CS-Ag NC) (**B**). The HR-TEM image, particle size distribution, zeta potential and XRD pattern analysis of CS-NPs and CS-Ag NC are shown as **I**–**IV** in each panel, respectively. The HR-TEM images of CS-NPs and CS-Ag NC were captured at 200 nm and 50 nm magnifications, respectively. The numbers in the images are showing the respective sizes of the CS-NPs and CS-Ag NCs.

**Figure 3 polymers-15-02961-f003:**
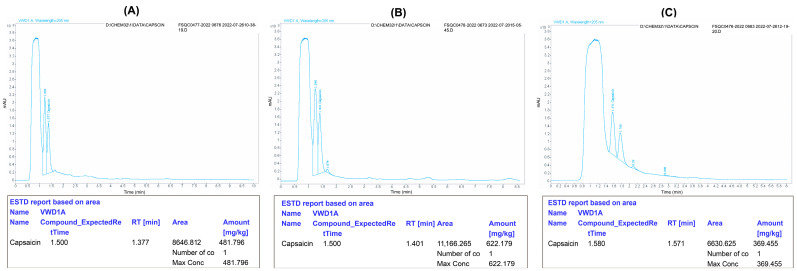
Effect of chitosan 200 ppm silver nanocomposites (CS-AgNC) in post-inoculation treatment on the capsaicin content in pepper pods. The capsaicin content in pepper pods of the treated plants (**A**) can be compared to the healthy control (**B**) and infected control (**C**). The original images were edited using Adobe Illustrator 2020 software for more clarity and have been provided as Appendix A.

**Table 1 polymers-15-02961-t001:** Differential hosts of the alfalfa mosaic virus (AMV) tested by mechanical inoculation.

Family	Test Plant	Common Name	Symptoms *	Molecular Detection	
				ELISA	RT-PCR
*Apocynaceae*	*Catharanthus roseus*	Periwinkle	SM&Y	0.502 ± 0.05	700 bp
*Chenopodiaceae*	*C. amaranticolor*		CLL	0.503 ± 0.02	700 bp
	*C*. *quinoa*	Quinoa	CLL	0.501 ± 0.01	700 bp
	*C. murale*	Nettle-leaved Goosefoot	CLL	0.497 ± 0.03	700 bp
*Fabaceae*	*Vicia faba*	Faba bean	SN&D	0.494 ± 0.04	700 bp
	*Phaseolus vulgaris*	Bean	NLL&CF	0.483 ± 0.07	700 bp
	*Vigna ungiuculata*	Cowpea	NLL	0.490 ± 0.02	700 bp
*Lamiaceae*	*Ocimum basilicum*	basil	SM&Y	0.504 ± 0.07	700 bp
*Solanaceae*	*Datura stramonium*	Jimson weed	SM	0.489 ± 0.09	700 bp
	*Nicotiana tabacum*	Tobacco	SM	0.495 ± 0.08	700 bp

* SM: systemic mosaic, Y: yellowing, CLL: chlorotic local lesion, SN: systemic necrosis, D: death, NLL: necrotic local lesion and CF: chlorotic flecking.

**Table 2 polymers-15-02961-t002:** Effect of chitosan silver nanocomposites and chitosan nanoparticles on virus infectivity.

Concentrations	Pre-Inoculation	Post-Inoculation	Simultaneously with AMV Inoculation
	Inhibition %	Inhibition %	Inhibition %
CS-Ag NC			
200 ppm	67 a	91 a	78 a
150 ppm	64 ab	89 a	68 abc
100 ppm	57 bcd	82 b	64 bc
50 ppm	53 cd	69 c	59 d
CS-NPs			
400 ppm	60 abc	90 a	76 ab
200 ppm	54 cd	86 ab	65 bc
150 ppm	51 cd	81 b	63 bc
100 ppm	47 d	66 c	57 cd
Healthy control	0 e	0 d	0 d
Healthy control treated with CS-NPs	0 e	0 d	0 d
Healthy control treated with CS-Ag NC	0 e	0 d	0 d
Infected control	0 e	0 d	0 e
L.S.D. at 0.05	5.829	5.11	9.37

The statistical significance or non-significance among the treatments is indicated with different letters (a–e). The means in each column, followed by the same letter, were not significantly different according to Duncan’s multiple range test, *p* ≤ 0.05. Four plant seedlings were used for each treatment. LSD values at 0.05 were statistically significant.

**Table 3 polymers-15-02961-t003:** Effect of various concentrations of chitosan silver nanocomposites and chitosan nanoparticles on active ingredients in pepper pods.

		Pre-Inoculation		Post-Inoculation			Simultaneously with Virus Inoculation		
Concentrations	Phenol mg/100 gFW	Capsaicin mg/kg DW	Proline mg/g FW^−1^	Phenol mg/100 gFW	Capsaicn mg/kg DW	Proline mg/g FW^−1^	Phenol mg/100 gFW	Capsaicin mg/kg DW	Proline mg/g FW^−1^
CS-Ag NC									
200 ppm	1.62 a	401.20 b	1.14 a	1.83 a	481.79 a	1.23 a	1.75 a	472.33 bc	1.17 a
150 ppm	1.53 a	393.63 c	1.10 a	1.78 a	473.56 a	1.18 c	1.63 b	468.86 bc	1.11 b
100 ppm	1.47 ab	385.73 e	1.06 a	1.67 a	469.87 a	1.12 b	1.58 ab	459.73 b	1.08 c
50 ppm	1.38 ab	376.53 h	1.03 ab	1.55 a	458.97 a	1.09 f	1.49 abc	445.40 bc	1.04 de
CS-NPs									
400 ppm	1.58 a	390.73 d	1.13 a	1.80 a	478.83 b	1.20 b	1.70 a	470.47 bc	1.16 a
200 ppm	1.48 ab	382.63 f	1.01 ab	1.76 a	468.76 a	1.15 d	1.59 b	463.91 bc	1.07 cd
150 ppm	1.39 ab	378.52 g	0.99 ab	1.63 a	463.53 a	1.09 f	1.47 abc	451.78 bc	1.05 cde
100 ppm	1.29 ab	371.43 i	0.96 ab	1.52 a	453.81 a	1.05 g	1.39 abc	439.56 c	1.02 d
Healthy control	1.12 b	622.17 a	0.75 b	1.12 a	622.17 a	0.75 i	1.12 c	622.17 a	0.75 g
Healthy control treated with CS-NPs	1.51 a	623.06 a	0.83 c	1.55 b	627.09 b	0.87 e	1.53 d	625.11 a	0.85 e
Healthy control treated with CS-Ag NC	1.59 c	626.03 a	0.86 c	1.65 c	629.13 b	0.89 e	1.62 e	627.13 a	0.87 e
Infected control	1.22 ab	369.45 j	0.95 ab	1.22 a	369.45 a	0.95 h	1.22 c	369.45 d	0.95 f
L.S.D. at 0.05	0.25	1.76	0.19	0.47	86.5	0.015	0.26	32.7	0.027

Means in each column followed by the same letter are not significantly different, according to Duncan’s multiple range test, *p* ≤ 0.05. LSD values at 0.05 are statistically significant.

**Table 4 polymers-15-02961-t004:** Effect of chitosan silver nanocomposites and chitosan nanoparticles on the growth and yield of pepper plants inoculated with AMV.

		Pre-Inoculation			Post-Inoculation				Simultaneously with Virus Inoculation			
Concentrations of	Pl. Height (cm)	Pod Weigh (g)		N. Pod per Plant	Pl. Height (cm)	Pod Weigh (g)		N. Pod per Plant	Pl. Height (cm)	Pod Weigh (g)		N. Pod per Plant
CS-Ag NC		Fresh	Dry			Fresh	Dry			Fresh	Dry	
200 ppm	58.8 a	20.0 ab	9.9 ab	23.0 a	60.6 a	21.2 a	10.3 ab	24.7 a	59.5 a	20.8 a	9.9 bc	23.8 a
150 ppm	57.8 a	19.3 ab	9.1 ab	21.8 a	59.3 a	20.5 a	9.8 ab	23.4 a	58.2 a	19.9 a	9.2 cd	22.6 a
100 ppm	56.9 a	18.0 ab	7.9 ab	20.7 a	57.9 a	19.8 a	9.1 ab	22.6 a	57.9 a	18.4 a	8.3 cde	21.5 a
50 ppm	55.9 a	16.9 ab	6.8 b	20.0 a	56.8 a	18.5 a	8.6 ab	21.8 a	56.2 a	17.9 a	7.6 de	20.4 a
**CS-NPs**												
400 ppm	58.0 a	19.9 ab	9.1 ab	22.3 a	58.9 a	20.6 a	10.1 ab	23.5 a	58.9 a	20.1 a	10.2 bc	22.7 a
200 ppm	57.0 a	18.6 ab	8.0 ab	21.8 a	57.7 a	19.3 a	9.2 ab	22.9 a	57.8 a	19.2 a	9.0 cd	22.1 a
150 ppm	56.1 a	17.0 ab	7.0 b	21.3 a	56.5 a	18.7 a	8.8 ab	21.8 a	56.8 a	17.9 a	7.5 de	21.6 a
100 ppm	55.9 a	16.7 ab	6.9 b	20.7 a	56.2 a	17.5 a	7.3 b	21.3 a	56.0 a	17.1 a	6.8 e	20.9 a
Healthy control	55.8 a	16.1 ab	6.5 b	20.6 a	55.8 a	16.1 a	6.5 b	20.6 a	55.8 a	16.1 a	6.5 e	20.6 a
Healthy control treated with CS-NPs	60.3 a	22.1 a	11.0 ab	25.3 a	60.9 a	22.9 a	11.8 a	25.7 a	60.7 a	22.5 a	11.3 ab	25.5 a
Healthy control treated with CS-Ag NC	61.1 a	23.0 a	12.0 a	26.0 a	61.9 a	23.7 a	12.5 a	26.8 a	61.7 a	23.4 a	12.3 a	26.5 a
Infected control	43.6 b	8.7 b	2.3 c	11.8 b	43.6 b	8.7 b	2.3 c	11.8 b	43.6 b	8.7 b	2.3 f	11.8 b
L.S.D. at 0.05	11.6	7.57	2.89	3.94	9.1	5.13	2.57	3.7	7.6	11.83	1.53	6.5

Means in each column followed by the same letter are not significantly different, according to Duncan’s multiple range test, *p* ≤ 0.05. LSD values at 0.05 are statistically significant.

## Data Availability

Not applicable.

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
