# Peer review of "Antiviral Activity of Chitosan Nanoparticles and Chitosan Silver Nanocomposites against Alfalfa Mosaic Virus"

_polymers, 2023, doi:10.3390/polym15132961_

Round 1
Reviewer 1 Report
In the article by El-Graine et al. the effect of chitosan on the alfalfa mosaic virus infection in pepper plants was studied. Two types of chitosan preparations were used: chitosan nanoparticles (CS-NPs) and chitosan nanoparticles in combination with silver ions (CS-Ag NPs). Both drugs were applied to the leaves of experimental plants a day before inoculation with the virus, simultaneously with inoculation or a day after inoculation. The results presented in Table 2 show that all three treatment modes lead to a significant reduction in the number of infected plants. It is assumed that the antiviral activity of chitosan is due to the direct action of chitosan nanoparticles on viral particles within the infected cell, as a result of which viral particles are destroyed and lose their infectivity.
This work can be of great practical importance. Preventing infection of plants by viruses is the important task of applied virology, phytopathology, and agronomic practics. The treatment of plants with chitosan makes it possible to achieve this goal, as was first shown by Pospieszny (Pospieszny H., Chirkov S., Atabekov J. Induction of antiviral resistance in plants by chitosan. Plant Science, 1991, 79:63-68) and in a number of recent publications cited in the manuscript.
However, the data given in Table 2 raises several questions. First, it is necessary to explain in footnotes what the letters from 'a' to 'e' in the columns mean. Secondly, it is unclear how many plants were in each group: 4, as indicated in line 192, or 20, as indicated in line 317. In any case, it is unclear where such percentages as 91, 76, etc. came from. Thirdly, it is not specified how long after inoculation the virus infection was checked, how many times such a check was carried out and which leaves were tested - only treated with chitosan, or untreated (systemic) leaves too? By what method (ELISA, RT-PCR) was this analysis carried out?
The sizes of CS-NPs and CS-Ag NC were 30 and 11 nm (lines 270 – 273). It is doubtful that particles of this size can pass through the cell membrane. Perhaps another mechanism of the chitosan activity should be discussed, namely the induction of the antiviral reaction in the treated plants.
I think the manuscript can be published in Polymers, however, minor revision addressing the comments should be performed before acceptance.
Some misprints
Line 54: Fabaceae
Line 79: 100 mg/L
Line 87: studies on
Line 102: Loewe Biochemica
Table 1: + in the RT-PCR column should be removed
Reviewer 2 Report
Authors study in this manuscript the antiviral activity of chitosan nanoparticles and chitosan silver nanocomposites against the alfamovirus Alfalfa mosaic virus (AMV). They assess infectivity under different treatment conditions and quantify a series of parameters, such as levels of capsaicin or proline, to evaluate the efficiency of the various chitosan applications. I think the results are interesting and the work address a relevant topic that tries to find solutions to future agricultural challenges.
My main concern comes from the way the article is written and how the results and the discussion are presented. First, the subsections in results are quite small and in my opinion are lacking a proper and detail description. Some of the figures (2 and 4) are not mentioned in the text and figures and tables go one after another without any text in between that could clarify anything. Specifically, it seems that experiments corresponding to figures 1-4 and table 1, are all focused on the preparation of the AMV inoculum for testing. From my point of view, this is not the topic of the manuscript and authors should consider the possibility of merging all these figures and tables in a single figure and merging sections 3.1 and 3.2 in a single one. The data that are more trivial, like the PCR amplification shown in figure 4, could be shown as a supplementary material. Similar to what I just mentioned, I would merge figures 5 and 6 into a single figure. Except for the HR-TEM image shown in A, in the other cases data on the different particles can probably be shown using the same graphs and just changing the colors or something like that. In this way it would be easier to compare both results with different particles. Additionally, the numbers in both A parts of the figure are unreadable to me. Authors can consider also showing a zoom image to better observe the actual chitosan particles.
Following with the analysis of the results, I think table 2 is not very clear for presenting the infectivity results. First, the table is in two different pages, which makes it difficult to read. Second, it has all the statistical letters together with the actual numbers and that is also hard to separate and follow. Additionally, I do not see the advantage of showing in the same table inhibition and incidence, since these two parameters are inversely proportional and knowing one is enough to easily calculate the other. Besides, it is not clear to me how viral infection was calculated in each case. Was it based on symptomatology? ELISA tests? PCR o qPCR amplification? Part 3.5 is also very short. Probably can be merged to 3.4 to give it more relevance. In any case, How do authors know that the white circles shown in A correspond to AMV particles? Are there any equivalent particles in the other samples? If not, does this indicate that all viral particles have been modified by chitosan? This could be included in the discussion. In addition, letters and numbers below the images are unreadable. Same things occurs in Figure 8. I can barely understand anything because I cannot read the numbers or understand the words in the different images.
In the discussion, I think a large part of it actually corresponds to the results section (lines 405-425 or lines 430-437). Additionally, there is a previous work that authors mentioned in the introduction and in line 452 that is quite relevant to this work because it basically tests similar particles against the same virus in a different plant species. The result theses authors obtained is similar in terms of antiviral activity, but there are interesting differences, like optimal concentration or application time, that authors fail to mention and I believe are important for discussion.
Reviewer 3 Report
Dear Editor,
I am writing to express my sincere gratitude for inviting me to review the manuscript titled "Antiviral activity of chitosan nanoparticles and chitosan silver nanocomposites against alfalfa mosaic virus". It has been an honor to contribute to the peer review process for Polymers. The manuscript presents valuable insights into the antiviral potential of chitosan nanoparticles (CS-NPs) and chitosan silver nanocomposites (CS-Ag NC) against alfalfa mosaic virus (AMV) infection in pepper plants. However, I have two minor concerns that I believe should be addressed before publication. I have one specific concern regarding the image quality.Firstly, in the section describing the electron microscopy results, it would be helpful if the authors could clarify the scale bar in the TEM image.
I recommend the inclusion of XRD (X-ray diffraction) peaks in the manuscript. XRD analysis would provide valuable information about the crystallographic structure of the chitosan nanoparticles and chitosan silver nanocomposites, further strengthening the characterization of these materials. Once these minor revisions are addressed, I am confident that the manuscript will be ready for publication. The study's robust methodology, well-structured presentation of results, and the significance of the findings align with the scope and objectives of the journal. Thank you for considering my recommendation. I appreciate your attention to these minor revisions, and I look forward to reviewing the revised manuscript.
Yours sincerely,
Dear Editor,
I am writing to express my sincere gratitude for inviting me to review the manuscript titled "Antiviral activity of chitosan nanoparticles and chitosan silver nanocomposites against alfalfa mosaic virus". It has been an honor to contribute to the peer review process for Polymers. The manuscript presents valuable insights into the antiviral potential of chitosan nanoparticles (CS-NPs) and chitosan silver nanocomposites (CS-Ag NC) against alfalfa mosaic virus (AMV) infection in pepper plants. However, I have two minor concerns that I believe should be addressed before publication. I have one specific concern regarding the image quality.Firstly, in the section describing the electron microscopy results, it would be helpful if the authors could clarify the scale bar in the TEM image.
I recommend the inclusion of XRD (X-ray diffraction) peaks in the manuscript. XRD analysis would provide valuable information about the crystallographic structure of the chitosan nanoparticles and chitosan silver nanocomposites, further strengthening the characterization of these materials. Once these minor revisions are addressed, I am confident that the manuscript will be ready for publication. The study's robust methodology, well-structured presentation of results, and the significance of the findings align with the scope and objectives of the journal. Thank you for considering my recommendation. I appreciate your attention to these minor revisions, and I look forward to reviewing the revised manuscript.
Yours sincerely,
Round 2
Reviewer 2 Report
I think the authors did a good job addressing most of my concerns. Nevertheless, I still have to disagree with three things:
i) In Figure 2 they use two images of different magnification. That is not very elegant, but at least it should be clearly indicated in the text to avoid confusion.
ii) Authors explain in their response that viral infections shown in table 2 were calculated by ELISA tests. However, I could not find this explanation in the actual text in section 3.3. I think this needs to be specified.
iii) Letters and numbers in Figure 3, previously known as figure 8, are still unreadable. Authors justify this because they were auto-generated with a software. Nevertheless, I think showing the numbers and letters as they are is useless. In my opinion, a possible solution is to edit the images stating clearly that they were edited and specifying that the actual numbers and data come from the mentioned software. For further clarity, the actual images obtained from the software can be included as supplementary material in a larger, more clear, format.
Author Response
The authors are grateful to the respected reviewer for his/her keen observation to further improve the manuscript. Each comment from reviewer 2 is followed by our replies.
Reviewer 2:
I think the authors did a good job addressing most of my concerns. Nevertheless, I still have to disagree with three things:
- i) In Figure 2 they use two images of different magnification. That is not very elegant, but at least it should be clearly indicated in the text to avoid confusion.
Reply I: The suggested changes have been done in the figure legends of Figure 2 (Lines 255-256).
- ii) Authors explain in their response that viral infections shown in table 2 were calculated by ELISA tests. However, I could not find this explanation in the actual text in section 3.3. I think this needs to be specified.
Reply II: The required information has been mentioned appropriately (Lines 191 and 260).
iii) Letters and numbers in Figure 3, previously known as figure 8, are still unreadable. Authors justify this because they were auto-generated with a software. Nevertheless, I think showing the numbers and letters as they are is useless. In my opinion, a possible solution is to edit the images stating clearly that they were edited and specifying that the actual numbers and data come from the mentioned software. For further clarity, the actual images obtained from the software can be included as supplementary material in a larger, more clear, format.
Reply III: Figure 3 has been edited using Adobe Illustrator software to clarify the image information. Following the respected reviewer’s suggestions, the original images have been provided as supplementary material. The figure legends have been revised, accordingly (Lines 306-308).

Reviewer 3 Report
Dear Editor,
I recommend accepting this article for publication in the journal, as it aligns well with the scope and objectives of the journal.
Thank you for considering this recommendation.
Best regards,